# Genetic Variation and Sensory Perception of a Pediatric Formulation of Ibuprofen: Can a Medicine Taste Too Good for Some?

**DOI:** 10.3390/ijms241713050

**Published:** 2023-08-22

**Authors:** Julie A. Mennella, Mengyuan Kan, Elizabeth D. Lowenthal, Luis R. Saraiva, Joel D. Mainland, Blanca E. Himes, M. Yanina Pepino

**Affiliations:** 1Monell Chemical Senses Center, Philadelphia, PA 19104, USA; saraivalmr@gmail.com (L.R.S.); jmainland@monell.org (J.D.M.); 2Department of Biostatistics, Epidemiology and Informatics, University of Pennsylvania, Philadelphia, PA 19104, USA; bhimes@pennmedicine.upenn.edu; 3Department of Pediatrics, Perelman School of Medicine, University of Pennsylvania, Philadelphia, PA 19104, USA; lowenthale@chop.edu; 4Sidra Medicine, Doha P.O. Box 26999, Qatar; 5College of Health and Life Sciences, Hamad Bin Khalifa University, Doha P.O. Box 34110, Qatar; 6Department of Neuroscience, University of Pennsylvania, Philadelphia, PA 19104, USA; 7Department of Food Science and Human Nutrition and Department of Biomedical and Translational Sciences, University of Illinois at Urbana-Champaign, Urbana, IL 61801, USA; ypepino@illinois.edu

**Keywords:** genetic ancestry, ibuprofen, pediatric formulations, taste, irritation, chemesthesis, single nucleotide polymorphisms

## Abstract

There is wide variation in how individuals perceive the chemosensory attributes of liquid formulations of ibuprofen, encompassing both adults and children. To understand personal variation in the taste and chemesthesis properties of this medicine, and how to measure it, our first scientific strategy centered on utilizing trained adult panelists, due to the complex and time-consuming psychophysical tasks needed at this initial stage. We conducted a double-blind cohort study in which panelists underwent whole-genome-wide genotyping and psychophysically evaluated an over-the-counter pediatric medicine containing ibuprofen. Associations between sensory phenotypes and genetic variation near/within irritant and taste receptor genes were determined. Panelists who experienced the urge to cough or throat sensations found the medicine less palatable and sweet, and more irritating. Perceptions varied with genetic ancestry; panelists of African genetic ancestry had fewer chemesthetic sensations, rating the medicine sweeter, less irritating, and more palatable than did those of European genetic ancestry. We discovered a novel association between *TRPA1* rs11988795 and tingling sensations, independent of ancestry. We also determined for the first time that just tasting the medicine allowed predictions of perceptions after swallowing, simplifying future psychophysical studies on diverse populations of different age groups needed to understand genetic, cultural–dietary, and epigenetic factors that influence individual perceptions of palatability and, in turn, adherence and the risk of accidental ingestion.

## 1. Introduction

How medications are delivered often differs between children and adults. While adults typically take medicine as solid formulations, encapsulating unpleasant-tasting active pharmaceutical ingredients (APIs), young children, unable or unwilling to swallow pills or capsules, are treated with liquid formulations that contain sugars, salts, and flavor volatiles to improve palatability [1,2,3,4]. There are cultural exceptions to this generalization, however.

In the United Kingdom, a popular cold and flu medicine for adults is formulated as a powder, containing APIs and excipients such as lemon flavors and sweeteners [5], dissolved in hot water and served as a tea. When the manufacturer replaced the API paracetamol with ibuprofen in the 1990s, some consumers complained that the tea tasted “bad” [6,7]. The accuracy of these complaints was later confirmed by psychophysical studies on adults who rated aqueous ibuprofen solutions as tasting irritating and bitter after swallowing, and felt chemesthetic sensations (i.e., chemically evoked irritation, e.g., the “tingle” of carbonation) in the oral cavity and throat [8]. Current formulations of this medicine do not contain ibuprofen, confirming that individual differences exist in the taste and chemesthetic properties of ibuprofen and the importance of the palatability of medicines among adults.

For pediatric populations, ibuprofen is often delivered as a sweetened suspension because sweetness is not only a preferred taste but also effective in masking unpleasant tastes [9]. Over-the-counter (OTC) sweetened ibuprofen suspensions are among the most widely used pharmaceutics for children [10], yet a recent study shows that, like adults, some children experienced “tingling” and “burning” sensations in mouth and throat after swallowing an ibuprofen-containing chewable capsule, whereas others did not [11]. Of importance, ibuprofen accounts for most unintentional poison exposures among children under 6 years in the United States [12,13]. While child-resistant packaging has been shown to prevent pediatric pharmaceutic poisonings, unsupervised ingestion can occur when such packaging is disabled, the medicine is within the child’s reach, and the child thinks the medicine tastes good [14]. Taken together, these data led us to hypothesize that, although sweetened suspensions of ibuprofen taste bad enough to deter some from ingesting large volumes, those children who do not experience its irritant properties may find that the medicine ‘tastes like candy’ [15], thereby increasing their risks of overconsumption due to children’s inborn proclivity for sweetness [16].

The primary goal of the present study was (1) to begin addressing this hypothesis by (a) determining the degree of personal variation in the palatability of a popular over-the-counter (OTC) sweetened formulation of ibuprofen via detailed measurements of chemosensory perceptions in a trained adult sensory panel and (b) comparing these results with panelists’ genetics, by conducting a candidate gene association study on selected variants in/near sweet and bitter taste receptors (TAS1Rs and TAS2Rs) and related chemosensory transient receptor potential (TRP) channels. For this initial study, we did not test children because the psychophysical tasks needed were complex and lengthy, and because, as reported above, variation in the perception of ibuprofen exists in both adults and children [6]. Using traditional psychophysical methods found in basic research [17] and sensory evaluation research [6,18], we studied a panel of adults trained in the detailed sensory methods used. As a secondary goal, (2) we addressed the need recognized by regulatory agencies (e.g., European Medicine Agencies; FDA) [19,20] and pharmaceutical companies [21] for safe and simpler standardized measures of medicine palatability and determined for the first time that just tasting the medicine led to predictions of perceptions after swallowing, simplifying future testing.

## 2. Results

### 2.1. Panel Characteristics

The 154 panelists (64 males; 90 females) who met the inclusion criteria had a mean age of 34 ± 1 years and represented the diversity of the city where they lived: Philadelphia, PA, USA [22] (Appendix A). The genetic ancestry of the unrelated panelists (*n* = 141) was determined to be as follows: African (*n* = 63), European (*n* = 51), South Asian (*n* = 13), East Asian (*n* = 7), or American (*n* = 7; Appendix A). To assess response reliability, we repeated testing for 114 panelists 46 ± 1 days later. Psychophysical ratings of sweetness (r = 0.56), irritation (r = 0.50), bitterness (r = 0.50), and hedonics (r = 0.43) were significantly correlated between the two sessions (*p*’s < 0.001).

### 2.2. Chemosensory Phenotypes

As shown in Table 1, panelists rated the medicine as tasting less sweet and more irritating, bitter, and unpalatable after swallowing than after the sip-and-spit condition. Irritation ratings persisted and were higher in the delay condition (5 min after swallowing) versus the sip-and-spit condition, confirming an aftertaste effect.

Psychophysical ratings varied widely across panelists for all conditions. Sip-and-spit ratings for sweetness, irritation, bitterness, and palatability were predictors of ratings after swallowing (Figure 1A–D). Sweetness ratings varied greatly among panelists, from 1 (no sensation) to 98 (strongest imaginable), as did irritation (0 to 57, very strong), bitterness (0 to 51, very strong), and hedonic ratings (−64, strong dislike, to 98, strongest imaginable like). How much panelists liked the taste of the ibuprofen formulation was positively related to the hedonic ratings of its sweetening excipients, Ace-K (r = 0.37, *p* < 0.001) and sucrose (r = 0.37; *p* < 0.001), and negatively related to its irritation ratings (r = −0.30; *p* < 0.001).

A greater proportion of panelists experienced the urge to cough and throat tingling/scratching in the swallow condition than in the sip-and-spit condition, and had these sensations localized in the epiglottis, larynx, trachea, hypopharynx, and hypopharynx areas (Figure 1E–G). Over two-thirds (72%) experienced at least one of the eight sensations, and one-third (31%) coughed after swallowing. Overall, chemesthetic perception predicted palatability; for those who experienced no chemesthetic sensations (*n* = 42), palatability was positively correlated only with sweetness (r = 0.37; *p* = 0.02); for those who experienced one or more irritation sensations (*n* = 112), palatability was negatively correlated with irritation (r = −0.28; *p* < 0.001) and only slightly correlated with sweetness (r = 0.19; *p* = 0.045).

### 2.3. Genetic Associations with Chemosensory Phenotypes

When grouped by chemesthetic responses, panelists who had the urge to cough or felt scratchiness in the throat (43%, 66/154) rated the medicine as more irritating, less sweet, and less palatable (Figure 2A). This pattern was held in those of African genetic ancestry (Figure 2B), but those of European genetic ancestry differed only in sweetness and palatability patterns (Figure 2C). Panelists who experienced tingling sensations in the throat (35%, 54/154) perceived the medicines as more irritating and less palatable (Figure 2D); among panelists of African ancestry this pattern held (Figure 2E), but among those of European ancestry tingling was associated only with greater irritation (Figure 2F).

Genetic ancestry covariates PC1 and PC2 were associated with ratings of sweetness, palatability, and irritation (Appendix A) and the urge to cough or scratchiness or tingling in the throat (Appendix A). More panelists with European genetic ancestry, versus those with African genetic ancestry, experienced the urge to cough or throat scratchiness (59% vs. 32%; *p* = 0.004) and tingling sensations (45% vs. 24%; *p* = 0.03) after swallowing the medicine (Figure 2G). Compared to panelists of European genetic ancestry (*n* = 51), those of African genetic ancestry (*n* = 63) rated the formulation as sweeter (26.1 ± 2.2 vs. 36.7 ± 2.8; *p* = 0.005) and more palatable (2.4 ± 2.7 vs. 14.0 ± 4.1; *p* = 0.03).

Table 2 summarizes SNP–phenotype associations with significant or marginal significance (*p* < 0.10). Four novel associations were found between selected TRP family SNPs and sweetness ratings or chemesthetic sensations after swallowing, and five known associations were confirmed between taste receptor SNPs and the sweetness ratings of Ace-K or bitterness ratings of the control drug PTU. No marginal associations were found for other SNPs (*TAS2R10* rs597468, *TAS2R14* rs1015443, *TAS2R46* rs2708380, and *TRPM5* rs230169).

Further linear or logistic regression models on data of unrelated individuals revealed that minor allele A of *TRPM8* rs7593557 was associated with increased ratings of sweetness and a smaller percentage of panelists who experienced the urge to cough or scratchiness. However, when adjusted for genetic ancestry, the SNP–phenotype associations were no longer significant (Table 2). The A allele of *TRPM8* rs7593557 is rare in individuals of European genetic ancestry but a major allele in those of African genetic ancestry. For the two TRP SNPs that were statistically associated with experiencing tingling sensations in Fisher’s exact tests, logistic regression results found that minor allele A of *TRPV1* rs224534 did not change the likelihood of experiencing tingling sensations, while minor allele T of *TRPA1* rs11988795 significantly increased the likelihood of experiencing tingling sensations by 1.88 times, independent of genetic ancestry.

The established associations of bitter receptor SNPs such as *TAS2R38* rs713598 with the bitterness of PTU and those of *TAS2R9* rs3741845 and *TAS2R31* rs10845293 with the bitterness of Ace-K were confirmed and remained significant after adjusting for genetic ancestry (Table 2). The SNP rs35744813 near the sweet taste receptor gene *TAS1R3* was associated with the sweetness of Ace-K and sucrose, but its minor allele was not significantly associated with differences in perceived sweetness.

## 3. Discussion

We found reproducible individual differences among adults in the taste and palatability of a popular OTC ibuprofen-containing pediatric formulation and two of its sweetening excipients, and genetic variation contributed to some of these differences. Swallowing the formulation intensified perceived irritation and bitterness and diminished sweetness and palatability compared to the sip-and-spit condition. Chemesthetic sensations lingered in the throat, confirming prior reports of an aftertaste [8,11]. The most notable individual differences in perception were identified by grouping panelists by the chemesthetic sensations they experienced after swallowing; for those who experienced none of the eight checklist sensations, palatability related only to sweetness ratings. However, for those who experienced the urge to cough and felt scratchiness in the throat, the medicine tasted more irritating, less sweet, and less palatable whereas for those who experienced tingling in the throat, the medicine tasted more irritating and less palatable. When compared to those of European genetic ancestry, those of African genetic ancestry were less likely to experience these chemesthetic sensations and the medicine tasted sweeter, less irritating, and more palatable.

While some evidence suggested that ibuprofen is a chemesthetic irritant, particularly in the throat [8,23], and can interact with TRP channels in peripheral sensory neurons [24,25], this is the first investigation of its phenotype–genotype associations in humans. Our findings highlight the importance of accounting for genetic ancestry in this type of research [26]. While self-reported race is widely used as a proxy for genetic ancestry [27,28], genome-wide genotyping enabled us to determine individuals’ genetic ancestry using a genome-wide SNP set and to control for the effects of genetic ancestry that often confound these genetic associations. We identified novel associations between *TRPA1* rs11988795 and tingling sensations independent of genetic ancestry, suggesting that this SNP is a potential biomarker for ibuprofen-induced tingling sensations in diverse populations. TRPA1 has been reported to be selectively excited by ibuprofen in rodent sensory neurons [29], which provides functional evidence of its involvement in the response to ibuprofen. In contrast, we also identified a novel association between *TRPM8* rs7593557 and the urge to cough or the scratchiness sensations of ibuprofen, which was confounded by genetic ancestry, demonstrating the advantage of using genome-wide genotyping data to characterize genetic ancestry.

Trained in psychophysical methods, adult panelists rated an OTC sweetened formulation that contained the API ibuprofen and common sweetening excipients, providing a more complex flavor than do aqueous solutions of ibuprofen that are previously used [8]. While the excipients Ace-K and sucrose most likely masked some of the unwanted bitterness and irritation of the API [4], we found that individual perceptual differences in the sweetness of the formulation and of the excipient Ace-K alone were related to variation in a bitter *TAS2R9* receptor, a finding consistent with those of prior research [30]. Consequently, panelists’ ratings included not only variation in the taste of ibuprofen but also taste–taste and taste–irritation interactions of ibuprofen and excipients, and individual genetic variation. Both bitterness and chemical irritation can suppress sweetness, and vice versa [31,32]. Further, these data suggest that evaluating aqueous solutions of the API alone may not inform on either the acceptance or risk of formulation misuse, especially for those that contain palatable sweetening agents and flavor volatiles.

The present study is not without limitations. Although we identified a novel association between *TRPA1* and tingling sensations of ibuprofen, and confirmed established associations between *TAS2R38* and the bitterness of PTU [33] and between *TAS2R9* and *TAS2R31* the bitterness of Ace-K [30,34], the investigation into genotype–phenotype associations was limited by a small sample size. Further, we cannot rule out that different gLMS ratings and chemesthetic sensations for ibuprofen between individuals of African and European genetic ancestry may be due to nongenetic factors, as self-identified race/ethnicity is highly correlated with genetic ancestry. Research has shown that Americans who self-reported as Black most preferred significantly higher levels of sweetness [35] and had higher daily intakes of added sugar [36] than those who self-reported as White. While our evidence suggests that adults with African genetic ancestry perceived the pediatric formulation of ibuprofen differently than did those of European genetic ancestry, future association studies on taste and palatability need to include children and other age groups to determine the relative contributions of cultural–dietary, developmental, genetic, and epigenetic pathways [37,38].

Other taste psychophysical studies conducted in adults, which allow for more complex taste ratings and longer testing duration, have revealed personal variation in taste, irritation, and the palatability of pediatric medicines, including the first-line treatment for young children infected with human immunodeficiency virus [39] and, in the present study, ibuprofen, one of the most widely used non-steroidal anti-inflammatory drugs. In light of our secondary goal, to address safe and simpler standardized measures of medicine palatability, we determined that just tasting the medicine was a predictor of perception after swallowing, suggesting that this as a safe, streamlined, standardized measure of the palatability of medicines for adult panelists [20,40]. The methods employed herein yielded consistent and reliable individual differences over time. Further, we determined whether adults deemed a popular OTC pediatric ibuprofen formulation pleasant or unpleasant derived primarily from the perception of irritation in the throat and from genetic ancestry. Because ratings of the swallowed medicine were predictive of ratings from tasting alone, this method can be used as an initial assessment tool to study individual differences in the taste of other medicines, particularly when swallowing is not allowed.

Needs for Pediatric Research. We caution that the data generated from trained adults are not generalizable to children, and we emphasize that fundamental challenges and the need for more research in children remain, including the need for validated methods to assess the palatability and chemesthetic properties of medicines among young children, who are prone to attention lapses, tend to answer questions in the affirmative, and have limited language capabilities [4]. Because of the pronounced age-related changes in taste preferences and sensitivity [41] and differences among genetic ancestry groups, pediatric research is needed to determine whether sensitivities to ibuprofen and other APIs change with age and which chemesthetic or taste sensations predict nonadherence to a medication regimen or decrease risks of overingestion when unsupervised [4]. Such research, however, needs to represent the diversity of the target patient population of the medicine and account for genetic ancestry [26,42].

While much attention has been paid to improving the taste of the medicine, since taste is the key factor in children’s acceptance and medication adherence [43], there has been a long-standing fear that a medicine may taste too good (e.g., like candy) to some children due to excipients such as sweeteners and palatable flavors [15]. Recall that palatability was positively correlated only with sweetness in those adult panelists who did not experience chemesthetic sensations after swallowing the medicine. In a recent study, 7- to 12-year-old children rated the palatability of a chewable capsule formulation of ibuprofen using a hedonic scale and were asked open-ended questions, including whether they felt sensations in the mouth or throat areas [11]. More than half reported tingling or burning sensations after swallowing the capsule(s), but the remaining 40% did not, suggesting susceptibility to overconsumption in this group. Although further research is needed to confirm this is indeed the case among young children, support for the existence of personal variations in perception among children—the hallmark of human perception—also come from a historical example of individual differences in the taste-related risks of pediatric poisoning and research on denatonium (Bitrex™). In the 1980s to early 2000s, legislation in some US states mandated the addition of this “highly unpalatable”, stable chemical to a variety of toxic liquids (e.g., paints, detergents, and antifreeze) to prevent poisonings and suicides. US poison control data, however, revealed that this bittering agent did not deter poisoning in all children (or adults), and thus its use could not be justified [44,45]. By taking the perspective of precision medicine that “nobody is average”, a greater understanding of the genetic and phenotypic variation underlying how a given medicine tastes to individual patients may inform clinical care and lead to different formulations tailored to their specific risks.

## 4. Materials and Methods

### 4.1. Participants and Stimuli

Healthy adults trained as sensory panelists were enrolled in a multisession study as trained sensory panelists to evaluate the taste of a variety of pediatric liquid formulations and excipients, one of which was an over-the-counter (OTC) sweetened suspension containing ibuprofen (100 mg/5 mL, 2% *w*/*v*) marketed for children (Berry-Flavored Children’s Motrin™ Oral Suspension, Johnson & Johnson Consumer Inc., McNeil Consumer Healthcare Division, Fort Washington, PA, USA) herein referred to as the ibuprofen formulation plus two of its excipients, 0.6 M sucrose and 0.12 M Acesulfame potassium (Ace-K), and 560 µM propylthiouracil (PTU) as a control stimulus for genotype–phenotype analysis [33]. Inclusion criteria encompassed healthy adults who were between 18 and 55 years old. Each adult was screened for medical eligibility by the study doctor (E.D.L.), who conducted a thorough review of all prescription and nonprescription medications and known allergies. Current smokers, those with known allergies or sensitivities to any of the medications or stimuli included in the taste panel, or who were taking or had taken in the recent past any medication that was contraindicated to those included in the taste panel were not eligible to participate. Each panelist who was a biological female took a urine pregnancy test prior to the start of each test session and was allowed to participate if test results were negative. Participants completed questionnaires to collect demographic information, including self-reported race. The Office of Regulatory Affairs at the University of Pennsylvania and the Institutional Review Board of the Children’s Hospital of Philadelphia approved all procedures and consent forms, and the study complied with the Declaration of Helsinki for medical research involving human Subjects. Each panelist gave written informed consent prior to screening. The trial was registered at clinicaltrials.gov (NCT013627351) prior to the start of the study (November 2018–July 2021).

The study was temporarily halted from March to July 2020 due to the COVID-19-related discontinuation of nonessential human subject research in Philadelphia. Upon study resumption in August 2020, we adapted the testing facility and received IRB approval to implement several procedures to reduce risks to study staff and participants. In addition to screening for COVID-19 symptomatology, we administered the National Institute of Health Olfaction Toolbox test [46] since anosmia or hyposmia could impact taste and hedonic ratings, and we did not retest the participants. As shown in Appendix B, we enrolled 169 adults of whom 15 were excluded because they failed the screening (*n* = 4) or did not understand psychophysical tasks/adhere to the protocol (*n* = 11). The self-reported races of the 154 panelists (64 males; 90 females) who met the inclusion criteria were as follows: 41% Black, 36% White, 13% Asian, 1% Native American, or 8% more than one race. The last 38 participants enrolled were tested during the COVID-19 pandemic. Although all were asymptomatic at the time of testing, 15% were categorized as having hyposmia or anosmia, an incidence above that reported prior to the pandemic [47]. We repeated the statistical analyses without these six panelists, and the findings remained unchanged.

### 4.2. Phenotyping

Using a double-blind study design, panelists were tested in closed rooms designed specifically for sensory testing, with red light illumination to eliminate the effect of the color of the solution, if any, on sensory ratings. During the first session, panelists were trained individually to identify basic taste sensations and were familiarized with rating taste and irritation sensations on the general labeled magnitude scale (gLMS) and palatability on the hedonic gLMS, on a computer with Compusense five™ Plus software (version 5) (Compusense, Inc., Guelph, Ontario, Canada). The gLMS visually spaces verbal descriptors of intensity for chemosensory perceptions (e.g., bitterness, sweetness, and irritation) as no sensation (0), barely detectable (1.4), weak (5.8), moderate (16), strong (35), very strong (53), and strongest imaginable (100) [48]. Because some individuals, despite training, confuse bitter with sour [49], we used the higher of the two ratings (hereafter referred to as bitterness), as carried out previously [50]. The hedonic gLMS ranked affective experience using the same visual arrangement, with the anchors strongest imaginable liking (100) and strongest imaginable disliking (−100) of any kind, with 0 indicating neither like nor dislike. Both scales rank perceived intensity along a vertical axis lined with adjectives spaced semi-logarithmically to yield ratio-quality data.

After swishing it in their mouth for 5 s, panelists rated taste, irritation, and palatability on a computer. One minute separated the presentation of each stimulus, during which panelists rinsed their mouths with water. Stimuli were presented in a quasi-randomized order since during the sessions in which they rated the ibuprofen medicine, it was presented last since it was the only stimulus that was swallowed. This formulation was evaluated under three conditions and in the following order: (a) swishing in mouth for 5 sec and spitting out without swallowing (sip-and-spit condition); (b) after a one-minute pause, swishing in the mouth for 5 sec and then swallowing (swallow condition); and (c) five minutes post-swallowing (delay condition). Immediately after rating the intensity of taste, irritation, and hedonics, panelists were presented with a checklist of eight chemesthetic sensations—burning, tingling, stinging, numbing, cooling, scratching, urge to cough, and urge to sneeze—and indicated which, if any, they experienced. They were then presented with a diagram of the sagittal section of the head and throat areas (i.e., tongue, hard palate, soft palate, oropharynx, nasopharynx, hypopharynx, larynx, trachea, esophagus, and epiglottis as shown in Figure 1G) and indicated where, if anywhere, they felt these chemesthetic sensations [6,29]. The investigator recorded if the panelist coughed or sneezed. This was repeated for each of the three conditions. The entire test session was repeated for a subset of participants to assess response reliability.

### 4.3. Genotyping, Genetic Ancestry, and Candidate Single-Nucleotide Polymorphisms (SNPs)

Saliva samples were collected from each enrolled panelist (*n* = 154). Eight samples that had poor DNA quality were not processed with a genotyping assay. Genotypes of the remaining individuals were assayed on Infinium Global Screening Array (Illumina, Inc., San Diego, CA, USA) and called using GenomeStudio (version 1.9). Quality control was performed at the variant and subject levels. At the variant level, 451,143 SNPs were obtained after applying the following criteria: (1) autosomal SNPs with a call rate of >95% across samples, and (2) a Hardy–Weinberg equilibrium *p*-value of > 10^−7^, and (3) with a minor allele frequency (MAF) of >0.01. At the individual level, samples with a genotype missing rate of >5% were removed; one sample failed to meet this criterion (Appendix A). We also determined whether or not individuals’ reported sex was consistent with their chromosomal sex by performing a “sex check” with PLINK (version 1.9) [51]. The reported biological sex of all individuals matched their chromosomal sex.

Kinship analysis was performed with KING (version 2.2.7) [52] to detect whether or not any panelists were relatives. Four pairs of related individuals were identified as having a first three-degree relationship, of which one individual from each pair (family) was retained, yielding a final sample size of 141 unrelated participants for genetic ancestry analysis (Appendix B). Genetic ancestry was assigned to study participants based on their overlap in the principal component (PC) space [53] created with genotypes of the 2504 reference subjects of the 1000 Genomes Project, which were labeled according to their super population clusters [54] as African (*n* = 661), European (*n* = 503), South Asian (*n* = 489), East Asian (*n* = 504), or American (*n* = 347). Specifically, the genotypes of common SNPs that were shared between study participants and those in 1000 Genomes Project were combined and PC analysis was performed on the combined panel with PLINK. Scatter plots of the first two principal components (PC1; PC2) were generated to project data from each panelist onto this lower-dimensional space and genetic ancestry was assigned to the 141 unrelated study participants, according to their overlap with the categories of the 1000 Genomes Project (Appendix A). Their genetic ancestry, which was determined to be African (*n* = 63), European (*n* = 51), South Asian (*n* = 13), East Asian (*n* = 7), or American (*n* = 7), was used in statistical analyses. Self-reported race was never used as a proxy for genetic ancestry.

The imputation of genotypes to a greater portion of the human genome was performed on Michigan Imputation Server (https://imputationserver.sph.umich.edu/index.html# (accessed on 21 January 2022)) with 1000 Genomes Project phase 3 v5 genotype data as a reference for unrelated individuals. Imputed variants were filtered based on the (1) imputation quality score of r^2^ > 0.30; (2) a Hardy–Weinberg equilibrium *p* value of >10^−7^, and (3) a MAF of > 0.01, yielding 12,072,402 imputed variants. To conduct the candidate gene association study, we selected SNPs within or near sweet and bitter taste receptors (TAS1Rs and TAS2Rs) and chemosensory transient receptor potential (TRP) channels that were biologically informed, including (a) rs35744813 near the sweet taste receptor gene *TAS1R3* [55]; rs3741845 within the bitter receptor gene *TAS2R9* [30]; and rs10845293 within the bitter receptor *TAS2R31* [34] since sucrose and Ace-K were excipients; (b) SNPs in the TRP family including rs230169 within the gene *TRPM5* (encoding TRP cation channel subfamily M member 5), the final step in the sweet taste receptor transduction cascade that leads to receptor depolarization [56]; rs11988795 within the gene *TRPA1*; rs7593557 within the gene *TRPM8*; and rs224534 within the gene *TRPV1* because ibuprofen is considered an irritant [57] and activates Trpa1 in rodent neurons [29]; (c) SNPs within other bitter taste receptors (*TAS2R10* rs597468, *TAS2R14* rs1015443, and *TAS2R46* rs2708380) that are considered generalists with 19 to 21 ligands each [58]; and (d) *TAS2R38* rs713598, which was used as an internal control for genotype-taste phenotype (PTC) associations [33]. For each SNP, we determined whether or not the distribution of alleles varied according to genetic ancestry.

### 4.4. Statistical Analyses

We computed descriptive statistics and used the Shapiro–Wilk test to assess for normality. Salty and savory ratings were rarely provided and therefore eliminated from the analysis. The gLMS ratings for sweetness were normally distributed (*p* > 0.20), but ratings of bitterness and irritation were not (Shapiro–Wilk normality test, *p* < 0.01). Square root transformation on original bitterness and irritation ratings (plus 0.01) was applied prior to analysis; however, the raw gLMS data were plotted to preserve the integrity of the meaning of the gLMS scale.

First, we focused on phenotyping data from 154 panelists. Separate ANOVA tests determined whether ratings intensified and the number and locations of sensations increased upon swallowing (sip-and-spit vs. swallow) and whether or not panelists experienced an aftertaste (sip-and-spit vs. delay), which was deemed to have occurred only when the taste or irritation ratings were greater in the delay condition than in the sip-and-spit condition. Separate correlational analyses were conducted to determine whether or not gLMS ratings in the sip-and-spit condition were predictive of ratings in the swallow condition. Dichotomous groupings were formed based on whether the individual experienced certain chemesthetic sensations or not and we determined whether or not the taste and palatability differed between groups using Fisher’s exact tests.

Second, we investigated the genetic associations with chemosensory phenotypes but analyses focused on the 141 unrelated individuals with successful genotyping and the determination of genetic ancestry (Appendix B). Separate ANOVA tests were conducted to determine whether or not candidate SNPs were associated with continuous, gLMS ratings, whereas Fisher’s exact tests were conducted to determine associations of SNPs with dichotomous groupings of sensory phenotypes. The relationship between genetic ancestry covariates (PC1 or PC2) and phenotypes was assessed using linear regression for continuous gLMS ratings (Appendix A) or logistic regression for dichotomous grouping (Appendix A). For SNP–phenotype relationships that showed marginal associations (*p* < 0.10), linear or logistic regression models for continuous or dichotomous phenotypes, respectively, without (non-adjusted) and with adjustments for genetic ancestry covariates (PC1; PC2), were used to determine whether or not identified associations were confounded by genetic ancestry [33]. Effect sizes were computed as (1) mean ± standard error of the mean (SEM) for continuous phenotypes, representing how many units of rating each copy of a minor allele changed, and the (2) odds ratio (OR) for dichotomous phenotypes, representing the increased risk of experiencing the sensation conferred by carrying one copy of the minor allele, along with the corresponding 95% confidence intervals (CIs). Comparisons between ancestry groups were limited to African (*n* = 63) and European (*n* = 51) due to the small sample size of other ancestry groups. Data were analyzed using R and Statistica (v13.1; StatSoft, Tulsa, OK, USA). The statistical significance level was set at *p* < 0.05 (two-tailed) for all analyses unless noted otherwise.

## Figures and Tables

**Figure 1 ijms-24-13050-f001:**
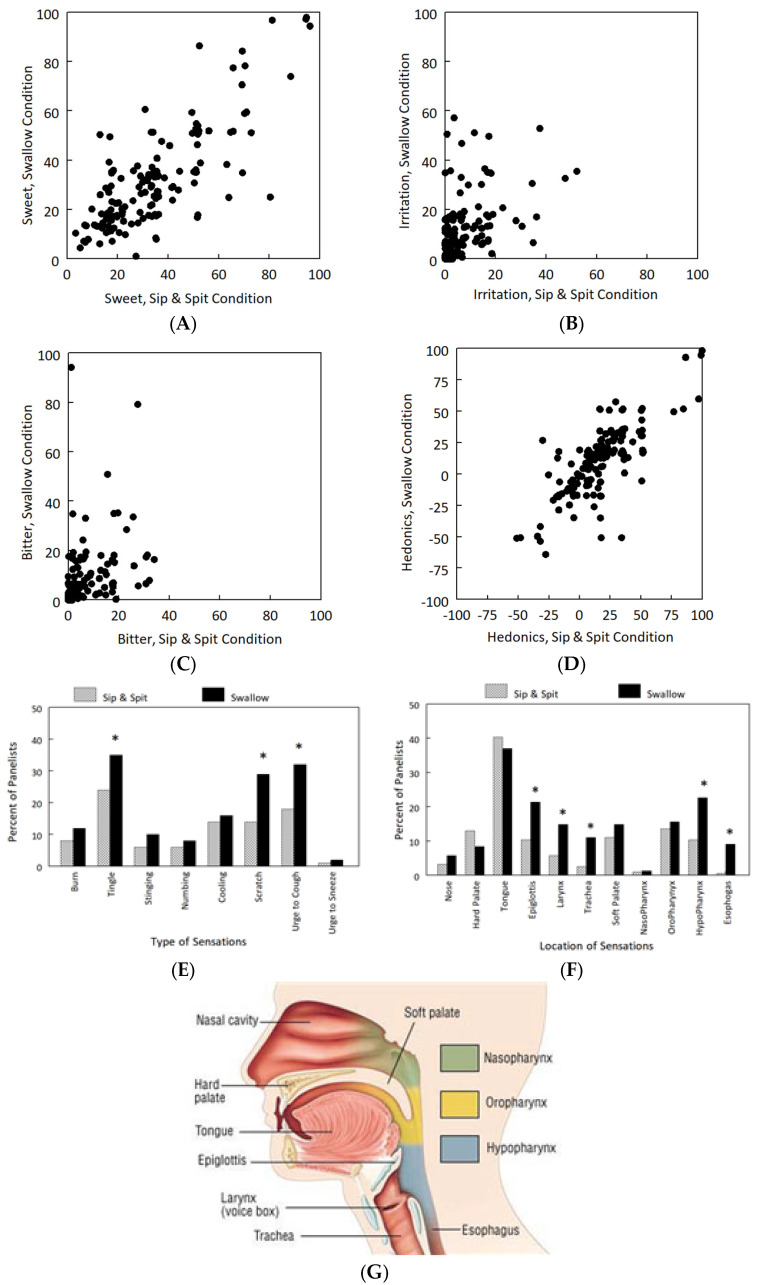
Sensory (gLMS) rating and chemesthetic sensations across experimental conditions (**A**–**G**). Relationships of gLMS ratings between sip-and-spit and swallow conditions for sweetness ((**A**); Pearson correlations: r = 0.78, *p* < 0.001), irritation ((**B**); r = 0.54, *p* < 0.001), bitterness ((**C**); r = 0.65, *p* < 0.001), and palatability (hedonics; (**D**); r = 0.76; *p* < 0.001) for all participants (*n* = 154). Each data point represents one panelist. Because bitterness and irritation ratings were not normally distributed, square root transformation was applied prior to analysis; however, original gLMS data are plotted to preserve the integrity of the meaning of the scale. (**E**–**G**) Percentages of panelists (*n* = 154) who experienced each of the listed chemesthetic sensations (**E**) and locations (**F**) during the spit-and-sip ((**F**), hatched bars) and swallow (solid bars) conditions. Panelists used the illustration of the sagittal section of head and throat areas (**G**) to indicate locations of sensations. * *p* < 0.05 compared to sip-and-spit condition, Fisher’s exact test.

**Figure 2 ijms-24-13050-f002:**
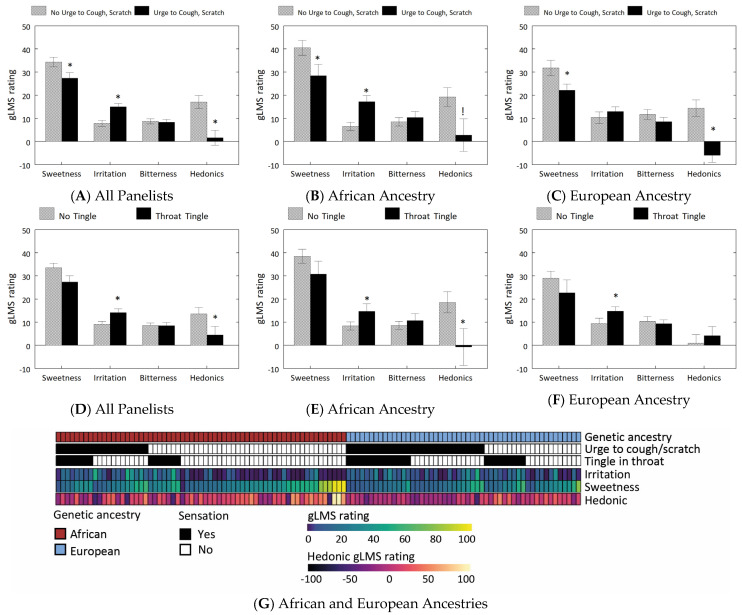
Relationships between sensory (gLMS) ratings, chemesthetic sensations, and genetic ancestry (**A**–**F**). gLMS ratings (mean ± SEM) for sweetness, irritation, bitterness, and hedonics of a pediatric ibuprofen formulation between panelists who did versus did not experience the urge to cough or scratchiness after swallowing (**A**–**C**) and between panelists who did and did not experience tingling in the throat (**D**–**F**). A and D, all panelists (*n* = 154). (**B**,**E**) those with African genetic ancestry (*n* = 63); (**C**,**F**) those with European genetic ancestry (*n* = 51). Because bitterness and irritation ratings were not normally distributed, square root transformation was applied prior to analysis; however, original gLMS data were plotted to preserve the integrity of the meaning of the scale. * Significant difference between ancestry groups (*p* < 0.05), ANOVA. Trend of difference (*p* = 0.06), ANOVA. (**G**). Heatmap of chemesthetic experiences (urge to cough/scratch; throat tingling) and gLMS ratings (irritation, sweetness, and hedonic) between individuals of African or European genetic ancestry.

**Table 1 ijms-24-13050-t001:** Effect of experimental condition on sweetness, bitterness, irritation, and palatability ratings of an ibuprofen-containing pediatric formulation ^1^.

		Experimental Condition		*p* Value ^2^	
	Sip & Spit	Swallow	Delay	Sip v Swallow	Sip v Delay
Intensity, gLMS rating					
Sweetness	33.9 ± 1.6	31.3 ± 1.6	16.4 ± 1.3	0.02	0.001
Irritation	6.7 ± 0.7	10.9 ± 1.0	8.9 ± 1.0	<0.001	0.02
Bitterness	7.3 ± 0.8	8.6 ± 0.8	6.6 ± 0.8	0.03	0.34
Palatability					
Hedonic gLMS rating	16.4 ± 2.1	10.4 ± 2.2	8.2 ± 2.1	<0.001	0.06

^1^ Over-the-counter ibuprofen formulation (Children’s Motrin™ Oral Suspension, Johnson & Johnson Consumer Inc., McNeil Consumer Healthcare Division). Because bitterness and irritation ratings were not normally distributed, square root transformation on the original data was applied prior to analysis; however, original gLMS data are summarized here to preserve the integrity of the meaning of the scale. ^2^
*p* values from one-way analysis of variance. Comparisons between sip-and-spit and swallow conditions determined how perceptions changed upon swallowing. Comparisons between sip-and-spit and delay conditions determined whether or not there was an aftertaste. To be considered an aftertaste, ratings had to be significantly higher in the delay than sip-and-spit condition. Data are least squared means ± SEM; *n* = 154.

**Table 2 ijms-24-13050-t002:** SNP—chemosensory phenotype associations of all ancestry groups (*n* = 141 unrelated individuals) and only those of African (*n* = 63) or European (*n* = 51) genetic ancestry.

Gene	SNP Identifier/ Function	Major/Minor Allele ^1^	Minor Allele Frequency			Phenotype, Stimulus ^2^	SNP-Phenotype *p* Value ^3^	SNP-Phenotype, Linear or Logistic Regression ^4^			
			All Ancestries	African Ancestry	European Ancestry			Nonadjusted		PC Adjusted	
								Effect Size	*p* Value	Effect Size	*p* Value
*TRPM8*	rs7593557/S419N	G/A	0.326	0.563	0.049	Sweet rating, Motrin	0.02	5.92 ± 2.15	0.007	3.49 ± 2.88	0.23
						Cough/scratch, Motrin	0.009	0.49 (0.30–0.77)	0.003	0.60 (0.32–1.10)	0.10
*TRPV1*	rs224534/T469I	G/A	0.28	0.143	0.324	Tingle, Motrin	0.04	—	0.23	—	0.96
*TRPA1*	rs11988795/Intronic	C/T	0.337	0.310	0.353	Tingle, Motrin	0.06	1.88 (1.12–3.23)	0.02	1.80 (1.05–3.15)	0.03
*TAS1R3*	rs35744813/Intergenic	C/T	0.369	0.619	0.127	Sweet rating, Ace-K	0.01	—	0.46	—	0.74
						Sweet rating, sucrose	0.07	—	0.18	—	0.93
*TAS2R9*	rs3741845/V187A	A/G	0.426	0.222	0.569	Bitter rating, Ace-K	<0.001	−0.99 ± 0.21	<0.001	−1.00 ± 0.25	<0.001
*TAS2R31*	rs10845293 A227V/	G/A	0.39	0.325	0.510	Bitter rating, Ace-K	0.002	0.69 ± 0.23	0.003	0.78 ± 0.23	0.001
*TAS2R38*	rs713598/A49P	C/G	0.465	0.508	0.412	Bitter rating, PTU	<0.001	1.72 ± 0.24	<0.001	1.73 ± 0.25	<0.001

^1^ Minor allele defined based on its frequency from all ancestry groups of the unrelated panelists (*n* = 141). ^2^ Sweet or bitter intensity ratings (continuous phenotypes) or the proportion of minor allele carriers who experienced chemesthetic sensations versus who did not (dichotomous phenotypes) after swallowing 5 mL of Children’s Motrin™ Oral Suspension, or ratings after tasting Ace-K, sucrose, and control drug PTU. Rating data for bitterness were square root transformed. ^3^
*p* values from one-way ANOVA (continuous phenotypes) or Fisher’s exact test (dichotomous phenotypes). ^4^ Linear (continuous phenotypes) or logistic (dichotomous phenotypes) regression model tested SNP–phenotype relationships in all unrelated panelists (*n* = 141). Effect sizes are the mean ± standard error of the mean for continuous phenotypes and odds ratios (95% confidence interval) for dichotomous phenotypes. *p* Values are from a regression model without (non-adjusted) and with PC1 and PC2 as covariates (PC adjusted). Dash (**—**) indicates effect sizes with *p* > 0.05. Abbreviations: Ace-K, Acesulfame potassium; PC, principal component; PTU, propylthiouracil; SNP, single-nucleotide polymorphism.

## Data Availability

Summary statistics for SNP–phenotype associations and R codes for statistical analysis are available upon reasonable request and the Data Usage Agreement should be sent to corresponding authors Julie A. Mennella (mennella@monell.org) or Mengyuan Kan (mengykan@pennmedicine.upenn.edu).

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
