# Peer review of "Genetic Variation and Sensory Perception of a Pediatric Formulation of Ibuprofen: Can a Medicine Taste Too Good for Some?"

_ijms, 2023, doi:10.3390/ijms241713050_

Round 1
Reviewer 1 Report
The overall Manuscript is well-designed and presented.
1. The conclusions drawn from the work are a little unclear. Precision medicine and bitter formulation cannot reduce accidental poisonings but may be difficult for the child to ingest the normal dose. The Conclusions can be modified for more clarity of the experimental design.
Reviewer 2 Report
This manuscript is titled "Genetic variation and sensory perception of a pediatric formulation of ibuprofen: Can a medicine taste too good for some?
My comments are listed below:
- Although this is an extremely well written manuscript, my biggest question/suggestion for the authors is to convince the readers about the problem statement. The Introduction section is brief and does not provide readers with enough context about the scientific need for this study.
- Ibuprofen suspension for children has been available for decades now without any serious concerns. The study uses Ibuprofen for Children but tests it on adults (who are not the consumers of this medicine) and then tries to generalize the study findings to children. This aspect of the manuscript is difficult for the readers to grasp. Why was this study not conducted in Children at the first place (describe in the manuscript in addition to what has been already written in the Discussion section; may be include as a separate sub-section)? Similarly, looking at the study findings it made more sense to conduct this study on adults using an adult formulation (for example: drug dextromethorphan). Authors must provide answers to these questions in their manuscript. In my opinion, the study objectives and study methodology are not fully in line.
Round 2
Reviewer 2 Report
Authors answered my comments satisfactorily.